# Dysembryogenetic Pathogenesis of Basal Cell Carcinoma: The Evidence to Date

**DOI:** 10.3390/ijms25158452

**Published:** 2024-08-02

**Authors:** Giovanni Nicoletti, Marco Saler, Umberto Moro, Angela Faga

**Affiliations:** 1Plastic and Reconstructive Surgery, Department of Clinical Surgical, Diagnostic and Pediatric Sciences, University of Pavia, Viale Camillo Golgi, 27100 Pavia, Italy; marco.saler@unipv.it; 2Advanced Technologies for Regenerative Medicine and Inductive Surgery Research Center, University of Pavia, Viale Brambilla, 74, 27100 Pavia, Italy; angela.faga@unipv.it; 3Surgery Unit, Azienda Socio-Sanitaria Territoriale di Pavia, Viale Repubblica, 34, 27100 Pavia, Italy; 4Integrated Unit of Experimental Surgery, Advanced Microsurgery and Regenerative Medicine, University of Pavia, Via Adolfo Ferrata, 9, 27100 Pavia, Italy; 5Independent Researcher, 14100 Asti, Italy; umberto.moro98@gmail.com

**Keywords:** basal cell carcinoma, embryology, pathogenesis, Hedgehog signalling pathway, embryogenesis

## Abstract

The Basal Cell Carcinoma (BCC) is a sort of unique tumour due to its combined peculiar histological features and clinical behaviour, such as the constant binary involvement of the epithelium and the stroma, the virtual absence of metastases and the predilection of specific anatomical sites for both onset and spread. A potential correlation between the onset of BCC and a dysembryogenetic process has long been hypothesised. A selective investigation of PubMed-indexed publications supporting this theory retrieved 64 selected articles published between 1901 and 2024. From our analysis of the literature review, five main research domains on the dysembryogenetic pathogenesis of BCC were identified: (1) The correlation between the topographic distribution of BCC and the macroscopic embryology, (2) the correlation between BCC and the microscopic embryology, (3) the genetic BCC, (4) the correlation between BCC and the hair follicle and (5) the correlation between BCC and the molecular embryology with a specific focus on the Hedgehog signalling pathway. A large amount of data from microscopic and molecular research consistently supports the hypothesis of a dysembryogenetic pathogenesis of BCC. Such evidence is promoting advances in the clinical management of this disease, with innovative targeted molecular therapies on an immune modulating basis being developed.

## 1. Introduction

Basal Cell Carcinoma (BCC) is the most common skin cancer worldwide and it accounts for over 85% of all non-melanoma skin cancers (NMSCs). Several risk factors are associated with this neoplasm and the development of BCC is due to a combination of extrinsic and intrinsic risk factors [1].

The main extrinsic risk factor is exposure to solar ultra-violet radiation (UVR) and its carcinogenic effect is dose-dependent. This is true not only of natural sunlight, but also of artificial UVR sources, such as indoor tanning devices. With regard to extrinsic risk factors, medical treatment options such as radiotherapy—which exposes both patients and medical staff to potentially harmful radiation—also play a role. Other than radiation, alcohol consumption, and iatrogenic immunosuppression in the context of organ transplantation, are extrinsic risk factors for BCC [1].

Intrinsic risk factors are old age, male gender and light complexion or low Fitzpatrick phototypes (I, II, III). A previous history of NMSC and immune system dysfunction, as in autoimmune diseases, can also result in a predisposition to BCC. Furthermore, genome-wide associated studies have identified over 30 loci associated with BCC susceptibility [1].

It is common knowledge that this tumour is highly insidious—for both its local invasiveness and its high incidence—in some highly functionally and aesthetically relevant anatomical sites.

According to European consensus-based interdisciplinary guidelines [2], the first-line treatment for BCC is radical surgical excision. Alternatively, photodynamic therapy can be considered as a treatment option for superficial and low-risk nodular BCCs. Radiotherapy represents a further valid option for patients who are not eligible for—or decline—surgery, and especially for elderly, fragile patients. Electrochemotherapy may be considered when surgery or radiotherapy are contraindicated. Patients with locally advanced and metastatic BCC should be offered Hedgehog pathway inhibitors (HHIs). A second-line treatment for patients with a progression of disease, a contraindication, or an intolerance to HHI therapy is immunotherapy with anti-PD1 (programmed cell death protein 1) antibodies (Cemiplimab).

The Basal Cell Carcinoma is a sort of unique tumour due to its combined peculiar histological features and clinical behaviour, such as the constant binary involvement of the epithelium and the stroma, the virtual absence of metastases and the predilection of specific anatomical sites for both onset and spread. Its incompletely explained nature justifies the different classifications that have been proposed over time for this neoplasm.

Molecular research over the last 30 years has shed light on the pathogenesis of BCC, highlighting the critical fundamental role of the Hedgehog (HH) signalling pathway—one of the key regulators of animal development, both during vertebrate embryonic development and in regulating adult stem cells involved in the maintenance and regeneration of adult tissues [3].

A potential correlation between the onset of BCC and a dysembryogenetic process has long been hypothesised. Tumour development is frequently described in the basic pathology literature as a recapitulation of embryogenesis.

In this literature review, we aim to highlight existing research from indexed publications that supports this theory.

Several review articles on the different pathogenetic hypotheses for BCC have been reported in the literature. However, to our knowledge, ours is the first comprehensive report on the overall evidence supporting the bond between BCC and embryology.

## 2. Materials and Methods

We searched for PubMed-indexed articles using the keyword “basal cell carcinoma” matching with “embryology”, “pathogenesis”, “embryo”, “embryogenesis” and ”hedgehog pathway inhibitor” in different combinations. The search strategy details are described in the Appendix A. A few articles in languages other than English (French, German and Polish) were included in this review—the authors being fluent in both French and German. The only retrieved article in Polish was translated for the authors by a Polish native speaker.

From the results of our search, we selected a total of 120 publications according to the aim of this review. The references from these selected articles were thoroughly scanned, too, thus obtaining a supplementary pool of 40 pertinent publications. This pool also included remote articles that were published before the advent of modern medical literature databases and that were missed by the PubMed search. Ultimately, 64 selected articles in total, with publication dates ranging from 1901 to 2024, were retrieved.

In addition to the conventional PubMed literature search, the query “Is there evidence supporting the embryological pathogenesis of Basal Cell Carcinoma?” was submitted to the OpenAI ChatGPT chatbot [4]. The generated text indicated that “The evidence supporting the embryological pathogenesis of Basal Cell Carcinoma (BCC) is still emerging, but there are several observations that suggest a connection between BCC and embryonic developmental processes:Hedgehog Signalling Pathway: Dysregulation of the Hedgehog signalling pathway, which is crucial in embryonic development, is a hallmark of BCC. Mutations in genes involved in this pathway, such as PTCH1 and SMO, are commonly found in BCC. These mutations can lead to uncontrolled cell growth and tumour formation, resembling the role of Hedgehog signalling in embryogenesis.Origin from Basal Cells: BCC arises from basal cells located in the basal layer of the epidermis. Basal cells are known to have stem cell-like properties and play a key role in embryonic skin development. The involvement of these basal cells in BCC suggests a link between BCC and embryological processes.Expression of Embryonic Markers: Studies have reported the expression of embryonic markers in BCC cells. For example, certain proteins or genes that are typically expressed during embryonic development have been found to be aberrantly expressed in BCC. This suggests a potential reactivation of embryonic programs in BCC pathogenesis.Tumour Microenvironment Resembling Developing Tissue: The microenvironment surrounding BCC shares similarities with the microenvironment of developing tissues. This includes factors such as altered extracellular matrix composition, signalling molecule gradients, and interactions with neighbouring cells, which can influence cancer growth and progression in ways reminiscent of embryonic development.Animal Models: Genetically engineered mouse models with mutations in Hedgehog pathway genes, or other embryonic signalling pathways, can develop BCC-like tumours. These models provide experimental evidence for the involvement of developmental pathways in BCC pathogenesis and support the embryological basis of BCC.

While these observations suggest a connection between BCC and embryonic development, further research is needed to fully elucidate the precise mechanisms underlying this association and its implications for BCC prevention and treatment.

## 3. Results

Our detailed analysis of the literature review identified a total of five main research domains on the dysembryogenetic pathogenesis of BCC (Table 1):(1)Correlation between the topographic distribution of BCC and the macroscopic embryology.(2)Correlation between BCC and the microscopic embryology.(3)The genetic BCC.(4)Correlation between BCC and the hair follicle.(5)Correlation between BCC and the molecular embryology.

The retrieved articles within the five research domains featured different volume and type profiles. The “Correlation between BCC and the molecular embryology” and “The genetic BCC” domains included thousands of reports consistently providing homogeneous and repetitive information. Therefore, only the most relevant articles were included in the review selection. Conversely, the other research domains featured a more composite literature profile; therefore, most of the retrieved articles were included in this review.

Interestingly, the further back in time the publication date, the easier it was to position the articles within the main research domains. By contrast, the more recent publications were spread across the literature within a broadly molecular research domain. Consequently, the retrieved articles were attributed to the different research domains according to their prevalent objective.

## 4. Discussion

### 4.1. Correlation between the Topographic Distribution of BCC and the Macroscopic Embryology

In 1952, Mohs and Lathrop first reported on the specific predilection of BCC for the embryonic fusion lines of the face, for both onset and spread [5]. They stated that BCC “arising over embryological fusion planes may invade to unexpected depth, owing, apparently, to the disposition of the condensed connective tissue in a direction perpendicular to the surface of the skin in these zones”.

In 1976, Jackson [6] revisited Mohs and Lathrop’s observations on the preferential topography of the onset and spread of BCC.

Later reports by other authors further highlight the specific tendency of BCC to spread and recur along and through the embryological fusion planes of the face.

In 1979 Panje reported on the predilection of the embryological fusion planes in a sample of 150 BCCs of the mid-face [7]. In 1986, Granstrom also demonstrated that 73 out of 120 BCCs were primarily located in a fusion line of the face [8]. On the other hand, in 2016, Armstrong [9] reported that in his experience, only 70 out of 331 BCCs (21.14%) were sitting in an embryological fusion plane of the face. Other similar publications, then, were based upon single case reports [10,11].

It is important to note that a lack of a definition of the specific anatomical sites corresponding to the embryological fusion planes of the face, was the main limiting factor in providing a systematic vision on this focus.

A true systematic vision on the correlation between the distribution of BCC and the embryological fusion planes of the face was provided by Newman and Leffell in 2007 [12]. They identified anatomical domains correlating to embryonic fusion planes in the midface, where BCC was four times more likely to occur than it was on other regions of the midface. From a total of 1457 facial tumours, 859 were located in the midface. Within this anatomical area, 35% of BCCs were located on the domain correlated to the embryonic fusion planes, accounting for 11.3% of the total surface of the midface. Therefore, the relative tumour density in the fusion plane domain was 3.06, compared to 0.74 for the lesions in the remaining sites of the face (*p* < 0.001). The large sample in this statistically sophisticated study, supported the possibility of an embryologic role for the pathogenesis of this neoplasm.

This hypothesis was critically reviewed in 2008 by Wentzell and Wisco [13], who highlighted that the main bias in Newman and Leffell’s study arose from the arbitrary identification of the number, site and size of the anatomical domains in the midface correlating to embryonic fusion planes. However, Wentzell and Wisco did agree that the study demonstrated BCC proclivity for some central facial cosmetic units. They also highlighted the need for a universal standardised designation for an “embryological fusion domain”. Only then could further research be conducted on the potential correlation of the process of embryological fusion with the onset and spread of BCC.

This debate highlighted the difficulty inherent in identifying in adult faces, the actual sites of fusion at the earlier embryological stage.

The question was settled in 2014 by Nicoletti et al. [14], who assumed that in the postnatal stage, the sites of congenital head and neck cleft malformations were likely to match the sites of fusion and/or merging of embryonic processes. Based on this assumption, the researchers correlated the distribution of BCC on the face and neck with the sites of congenital clefts of the head and neck using (1) the original anatomic diagram of the Tessier classification of craniofacial clefts [15] (Figure 1), (2) the anatomic diagram by Moore et al. [16] featuring the paths of the “hairline indicators” of craniofacial clefts that represent the cranial extensions of the Tessier classification (Figure 2), and (3) an anatomical diagram featuring the sites of congenital clefts of the neck (Figure 3). The Tessier classification of craniofacial clefts, subsequently completed by Moore et al., progressively numbered (from numbers 0 to 14, plus number 30) the paths of various congenital clefts of the face. In the absence of a universally accepted and coded mapping protocol, the researchers assigned locations to the congenital cleft malformations of the neck, using well-established sites along a line running from the chin to the clavicular notch in the anterior midline, that corresponded to the branchial arches on each side of the neck and to their anterior midline fusion line. The study demonstrated that the proportion of BCC localised within a cleft site was significantly higher than the 50% (frequency = 88.68%; 95% confidence interval = 86–100%, *p* < 1 × 10^−10^) one could expect to observe, by chance, in the same sites.

In 2018, the same research group analysed the distribution of BCC within the anatomical domain of the external ear—a site missed by both the Tessier and the Moore classifications of craniofacial clefts. The study demonstrated that the greatest number of BCCs in the ear was observed within the currently most-credited sites of the embryonic fusion planes of the auricle (Figure 4) [17]. In a sample of 72 BCCs deriving from 69 patients, 57 (79.17%, 95% CI  =  67.98–87.84) out of 72 neoplasms were localised within the embryonic fusion planes of the auricle. These sites displayed a 12-fold increased tumour incidence in comparison to the remaining surface of the ear.

The study of 2014 [14] was repeated in 2020 with the same methodology using a larger sample of BCC. It demonstrated a 93.10% correspondence of the topography of this neoplasm to a cleft site of the face and neck [18] vs. a 6.90% correspondence for non-embryologically relevant ones (*p* < 0.001).

### 4.2. Correlation between BCC and the Microscopic Embryology

In 1959, Pinkus [19] depicted BCC as a hamartoma that differentiates itself from other pure epithelial malignancies by its peculiar and constant close interaction between the epithelium and the stroma. He perceived this sort of organoid skin tumour as a monstrous attempt at skin adnexogenesis in the postnatal stage, through interaction of pathological ectodermal and mesodermal components which form fibroepithelial growths of varying degrees of maturity [20]. Other pure skin epithelial cancers, instead, derive from proper neoplastic transformation of a single cell and progress through a typical stroma-dissociated invasion. His first hypothesis regarding BCC was of an alternative pathogenetic model, one different from that of traditional carcinogenesis. Pinkus defined BCC as an aggressive tumour derived from any part of the equi-potential ectoderm of the skin in combination with organised mesodermal stroma. He considered this tumour the result of perverted and monstrous adnexal embryogenesis in the not fully competent adult skin, either in response to external carcinogenic stimuli or to innate causative factors. He finally described BCC as a fibro-epithelial unit closely related to benign adnexal tumours from which it might develop by progression. He further stated that, although aggressive, BCC is not fully malignant.

In 2018, Milosevic, using stem cell cultures [21], demonstrated a significantly higher expression of the embryonic markers Oct4, Sox2 and Nanog in neoplastic cells, compared to cells from the lesion margins and healthy tissue (*p* < 0.01). A higher expression of the mesenchymal markers CD44 and CD73 was also demonstrated in tumour cells compared to margin and control cells (*p* < 0.05). Only CD90 markers approximately displayed the same values in tumour cells and margin cells; these values were higher than in the control (*p* < 0.01). Bmi-1 was also over-expressed in tumour compared to tumour margin cells (*p* < 0.05). A considerably higher expression of GPR49 was appreciated in tumours vs. margin cells (*p* < 0.01) and control cells (*p* < 0.001).

### 4.3. Correlation between BCC and the Hair Follicle

The correlation between BCC and the development of the hair follicle is being deeply investigated and intensely debated, supported by modern genetic and molecular embryology research.

According to Pinkus’ theories on BCC—which were occupying his thoughts as early as 1938 [22,23]—this tumour resembles the primordial stage of cutaneous adnexa (pilary germ), in which undifferentiated foetal ectodermal and mesodermal cells form a unit [24].

This insight, on the microscopic morphological correspondence between BCC and the pluripotential and equipotential ectodermal pilary germ, was confirmed in 1978 by Kumakiri, who compared six BCCs to foetal and adult skin using electron microscopy [25].

Molecular similarities within foetal epithelial cells, adult cells in the hair follicle and infiltrating BCC cells were demonstrated by Airola in 1998 [26].

Sellheyer and Krahl, in 2008, stated the adnexal nature of BCC and suggested that BCC is the most primitive follicular tumour. This was due to the expression of the cell-cell adhesion molecule Ep-CAM—specifically, in the embryonic hair germ, the secondary hair germ of the terminal hair follicle and the outer root sheath of the vellus hair follicle [27].

Conversely in 2010, Youssef et al., using clonal analysis in a sample of 171 BCCs, demonstrated that 93% did not, as previously thought, originate from bulge stem cells. They did so from long-term resident progenitor cells of the interfollicular epidermis [28].

The same author later stated that the tumour-initiating cells are massively reprogrammed into a fate resembling that of embryonic hair follicle progenitors (EHFPs) [29].

Peterson et al., in 2015, identified additional hair follicle stem cell populations that are also susceptible to tumorigenesis [30]. These stem cells are specifically located in the upper bulge, lower bulge/secondary hair germ, isthmus and Touch Domes (TD)—innervated structures within the epidermis. They are not found in the interfollicular epidermis (IFE) stem cells or transit-amplifying matrix cells.

This topic is currently studied using molecular embryology methods, and the related selected research is presented in a dedicated section of this discussion.

### 4.4. The Genetic BCC

In 1960, in conjunction with the explosion of human genetic research and discoveries, further support for a dysembryogenetic pathogenesis of BCC was provided by Gorlin and Goltz. They reported on a genetically transmitted syndrome that consistently featured multiple BCC and a variety of birth defects, such as epidermal cysts, palmar and plantar pits, keratocysts of the jaw, calcified dural folds, various skeletal anomalies, cleft lip and/or palate and other neoplasms or hamartomas [31]. The inheritance of the syndrome is autosomal dominant. Therefore, a distinction was made between genetic BCCs and sporadic BCC, even though most BCCs are sporadic.

Modern molecular embryology has allowed the identification of the aspects that both genetic and sporadic BCC have in common. It would appear that the origin of these BCCs is related to a molecular disturbance of embryonic development induced by genetically inherited defects, or extrinsic risk factors [32,33].

### 4.5. Correlation between BCC and the Molecular Embryology

The HH gene, originally discovered in *Drosophila melanogaster*, plays a relevant role in embryogenesis across multiple mammal species—animal and human. The family of HH proteins consists of at least three different members: Sonic HH (SHH), Indian HH (IHH), and Desert HH (DHH). Even if this signalling pathway is implicated in the postembryonic regulation of stem-cell number in epithelia and blood, its activity seems to be reduced, or absent, in adult individuals [34]. The aberrant reactivation of the pathway is associated with the development of a number of human malignancies, including both inherited and sporadic BCC [35]. The pathway is finely tuned by a balance of transmembrane proteins, including those acting as activating factors, such as Smoothened (SMO), those with an inhibitory function, like Patched1 (PTCH1). The tumour suppressor gene PTCH, which is part of the HH signalling pathway, is mutated in both hereditary and sporadic BCC. The central role of the pathway, in both BCC formation as well as hair cycling, has been demonstrated. Given that BCC is supposed to be a follicular-driven tumour, it seems reasonable to infer that some element of fine tuning or alteration in the regulation of hair cycling could be the driving force towards the formation of benign follicular hamartomas, or, malignant BCC neoplasms [36,37,38,39].

The phenotype of HH/glioma-associated oncogene (Gli)-driven tumours might be determined by both the cell of origin and the tissue context (quiescent vs. growing hair follicles), with the development of superficial BCC from the interfollicular epidermis and nodular BCC from hair follicle stem cells [40,41].

Moreover, a bond amongst both genetic and sporadic BCC, the fine details in the embryonic hair follicle development, and wound healing does also exist [42,43,44,45,46].

It has also been demonstrated that the HH signalling pathway can be altered through a PTCH mutation and by UVR, thus confirming the traditional role of sun exposure in the development of BCC [47]. The biological relevant spectra of solar light include UVA (320–400 nm) and UVB (280–320 nm). The mechanisms of action of different UVR spectra are differential and, in some cases, overlapping [48]. UVB is absorbed by both the epidermis and the papillary dermis [49]. The occurrence of UVB signature mutations in key HH pathway genes in BCC has been demonstrated [50]. On the other hand, although UVA has 1000 times lower efficiency at inducing biological effects than UVB, it can reach the reticular dermis [49], thus leading to a strong reduction in the level of collagen 1 (Col1). It has been demonstrated that the decrease in Col1 expression is associated with the ability of SMO-M2-expressing cells to lead to BCC formation and invasion as well as to EHFPs to reprogram during tumour initiation [51].

Redirecting the fate of hair follicle progeny in the context of activated HH signalling might explain the association of BCC development not only with UVR exposure but also with tissue injury and chronic ulceration [52].

All of these factors might also explain the onset of BCC outside the embryological relevant sites on the face.

Anomalies in the SHH signalling pathway have also been shown to play a major role in craniofacial development. The survival of migrating neural crest cells, the fusion and/or merging of embryonic processes, and mutations in a number of pathway constituents all underlie craniofacial malformations [53,54,55,56,57,58,59]. After the injection of anti- SHH antibody (*p* > 0.05), developmental abnormalities have been demonstrated in chick heads, although with a weak statistical significance [60]. The correct balance of HH pathway signalling has been shown to be important in the fusion and formation of the face, and alterations in signalling lead to a variety of clefts. However, much still remains to be understood about the intricate tissue-specific and spatiotemporal regulation of Hedgehog pathway gene networks [61].

The Hedgehog pathway might also be responsible for the direct involvement of the nervous system in the development of BCC. In 2015, Peterson et al. demonstrated, in mice, that the formation of rare TD-derived lesions was attenuated in denervated skin (*p* = 0.04) [30]. The skin-sensitive nerve fibres express the Hedgehog ligand. This, in turn, triggers tumour development in upper and lower bulge stem cells of the hair follicle, producing follicular tumours in small inter-follicular cell clusters close to TDs, and leading to the subsequent development of proper BCCs. Considering the common neuro-ectodermal origin of both the epidermis and nerves, a mutually active interaction between these tissues might be expected in BCC, where nerves could be one element in the context of a combined organoid proliferation, as in a sort of triad: epithelium, stroma and nerve [62].

These considerations support the previously reported preferential distribution of BCC on the lines of Blaschko [63], the microscopy observations by Nödl [64] and Iyengar [65], and the evidence from experimental carcinogenesis in rabbits reported by Pawlowski [66].

In response to the central role of the HH signalling pathway in the development of BCC, a targeted immune therapy is currently under development [67,68,69]. At this point in time, the US Food and Drug Administration (FDA) has already approved two inhibitors of SMO for the treatment of advanced BCC: Vismodegib (GDC-0449; 30 January 2012) and Sonidegib (LDE-225; 24 July 2015).

Several other structurally distinct SMO inhibitors are also being evaluated, in both phase 1 and phase 2 clinical trials to treat locally advanced or metastatic BCC [70].

A significant aspect of novel BCC-focused therapies is the search for additional mutations, other than those related to the HH signalling pathway [71]. This genetic network of cancer-associated genes—seemingly more complex than that previously thought to be involved in BCC carcinogenesis—could have a potentially favourable impact on the development of new molecular targeted therapies [72].

At this stage, it seems appropriate to discuss the role of the OpenAI ChatGPT chatbot-based search used in this review article. The decision to use AI was taken in an effort not to miss any potentially relevant information on the topic. To our surprise, the data retrieved using AI tools did not provide novel information vs. that from the traditional literature search. Interestingly, even the PubMed-indexed articles failed to cover the whole body of knowledge on the specific topic of the dysembryogenetic pathogenesis of BCC. Some of the information was retrieved because our authors remembered having read certain relevant articles in the distant past. This historical memory relates to articles published before the advent of the modern biomedical bibliographic retrieval systems in 1964 [73].

The development of AI has greatly improved the potential for data analysis in biomedical sciences. As a tool, however, AI is still in its infancy and it will take time before its glorious potential can be harnessed to the full. The AI-generated answers to our query “is there evidence supporting the embryological pathogenesis of BCC” implied that the available observations only “suggest” a bond between BCC and embryology and that “further research is needed to fully elucidate” this hypothesis. Our search was ultimately based upon integrated knowledge from the authors’ individual, varied biomedical backgrounds: dermatology, oncology and malformative plastic surgery. We feel that “the suggestion” mooted in response to our AI question amounts to more than that and is, in fact, backed by concrete evidence. The pertinent information gathered is: (1) The same molecular signalling pathways are shared amongst embryogenesis, the genetic BCC and the development of the hair follicle and craniofacial congenital anomalies; (2) there is a histological morphological correlation between BCC and the hair follicle; (3) there is a statistically significant correlation between the sites of onset of BCC and the sites of craniofacial clefts; (4) the external agents (UVR, trauma, etc.) that, traditionally, are considered major aetiological factors for BCC, trigger the same molecular signalling pathways as above, and, therefore, they might be redefined as co-factors. We feel confident that our data can make a relevant contribution to the pool of information available for future AI-based searches. Any improvement in the quality of knowledge on this topic can only have positive implications for BCC prevention and treatment.

## 5. Limitations of the Study

The paper presented here has one fundamental limitation: Following its publication, as with every other literature review, it will inevitably become obsolete. Fortunately, in the particular area of study of our literature review, there exists a tumultuous and unstoppable surge of developments in molecular research. The resultant increase in knowledge regarding this most unusual of neoplasms will render our efforts in this endeavour worthwhile. We are confident that this literature review will be a useful launching pad for subsequent, in-depth studies into possible alternative solutions regarding the pathogenesis of BCC. Our search through the literature brought home to us the extraordinary availability of shared information that exists in the realm of English-language scientific resources. However, beyond this Anglocentric sphere, there might be sources from different cultural and linguistic backgrounds with advanced and valuable insights to contribute to the discussion. Only when we can all of us share and compare our findings with each other will we have a real strategy to advance knowledge of the true pathogenesis, and subsequently, the most appropriate treatment, of BCC.

## 6. Conclusions

A large amount of data from microscopic and molecular research consistently supports the hypothesis of a dysembryogenetic pathogenesis of BCC. However, gaps in the chain of evidence still persist in the field of topographic embryology, as accurate data on the distribution of BCC outside the head and neck domain are missing. More valuable information on the bond between perturbed embryogenesis and skin cancerogenesis can certainly be discovered. We believe the focus should be on the correlation of the topographic distribution of BCC and the embryological development of the body.

Evidence of dysembryogenetic pathogenesis in BCC has more than a purely speculative value. It could help promote advances in the clinical management of this disease—both in the prevention of the disease, by highlighting definite risk anatomical sites for this neoplasm, and for suggesting innovative targeted molecular therapies.

A novel field of investigation to undertake might be one providing a comprehensive view of disease-related genetic patterns, including both craniofacial malformations and skin cancer [74,75].

## Figures and Tables

**Figure 1 ijms-25-08452-f001:**
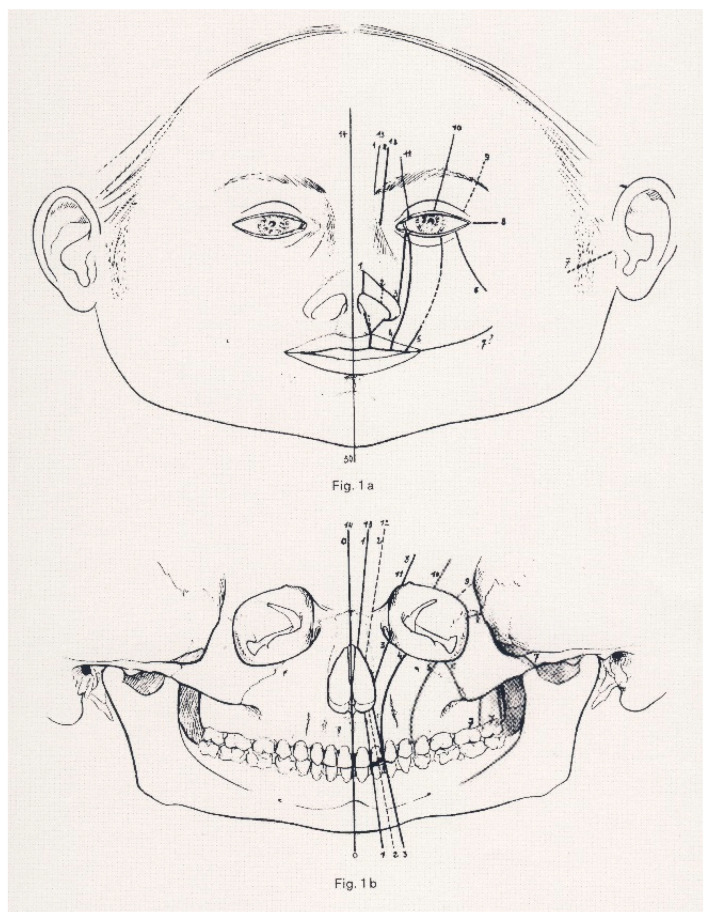
The original Tessier anatomical diagram of craniofacial clefts: localization on the soft tissues (**a**) and skeleton (**b**). Dotted lines are either uncertain localizations or uncertain clefts. Reprinted with permission from Ref. [15]. 1976, Elsevier.

**Figure 2 ijms-25-08452-f002:**
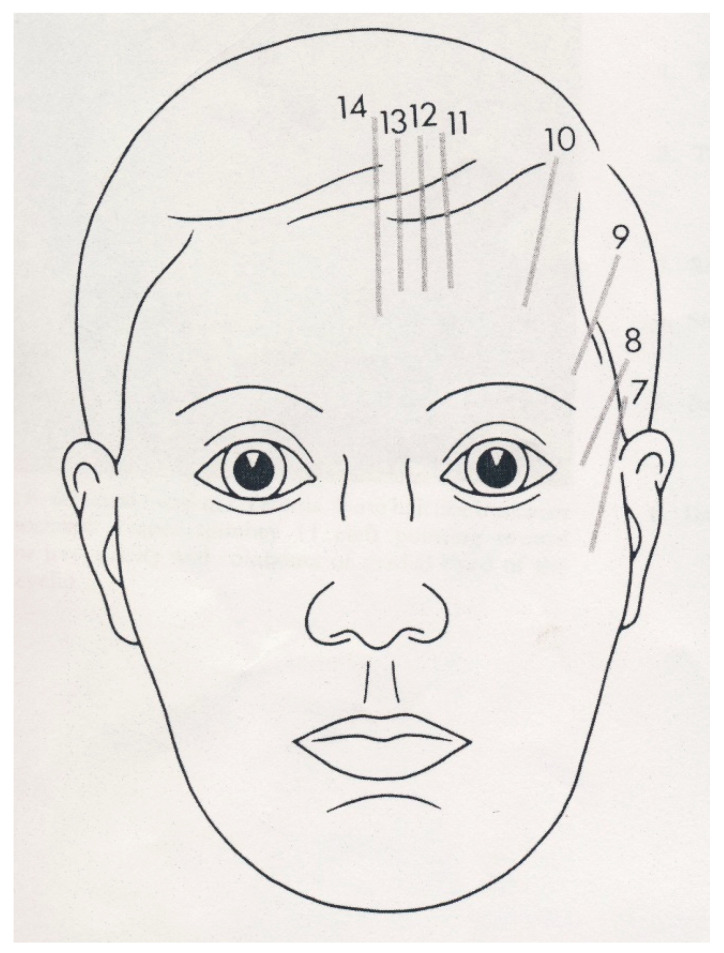
The hairline indicators are the superior and lateral extensions of the Tessier original craniofacial cleft classification. Reprinted with permission from Ref. [16]. 1988, Wolters Kluwer Health.

**Figure 3 ijms-25-08452-f003:**
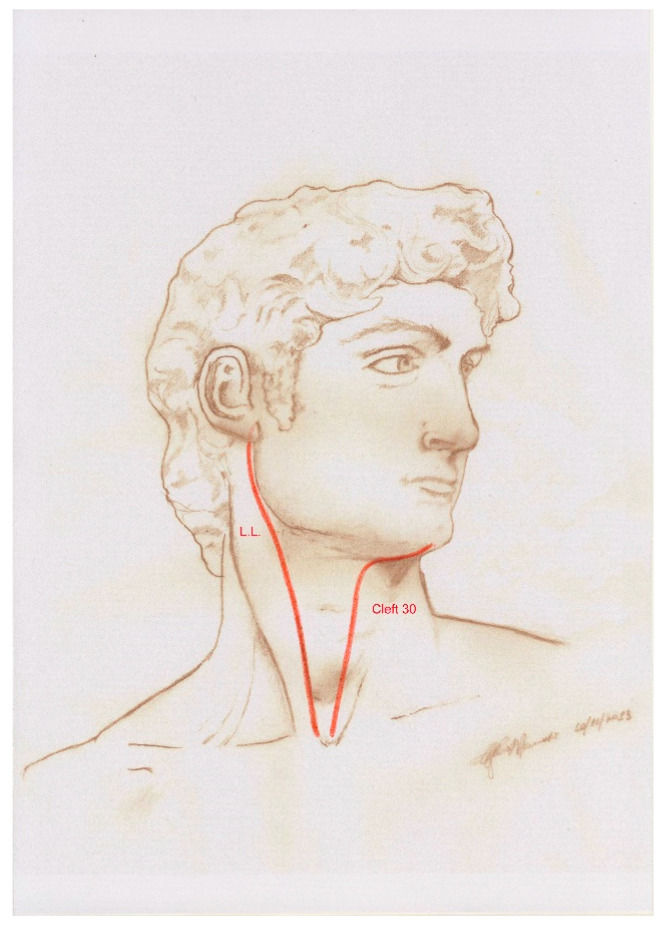
Anatomical diagram of the typical sites of congenital clefts, fistulas, and cysts of the neck: the laterocervical line (L.L.) and the anterior neck midline (Tessier cleft number 30). Reprinted with permission from Ref. [14]. 2014, Wolters Kluwer Health.

**Figure 4 ijms-25-08452-f004:**
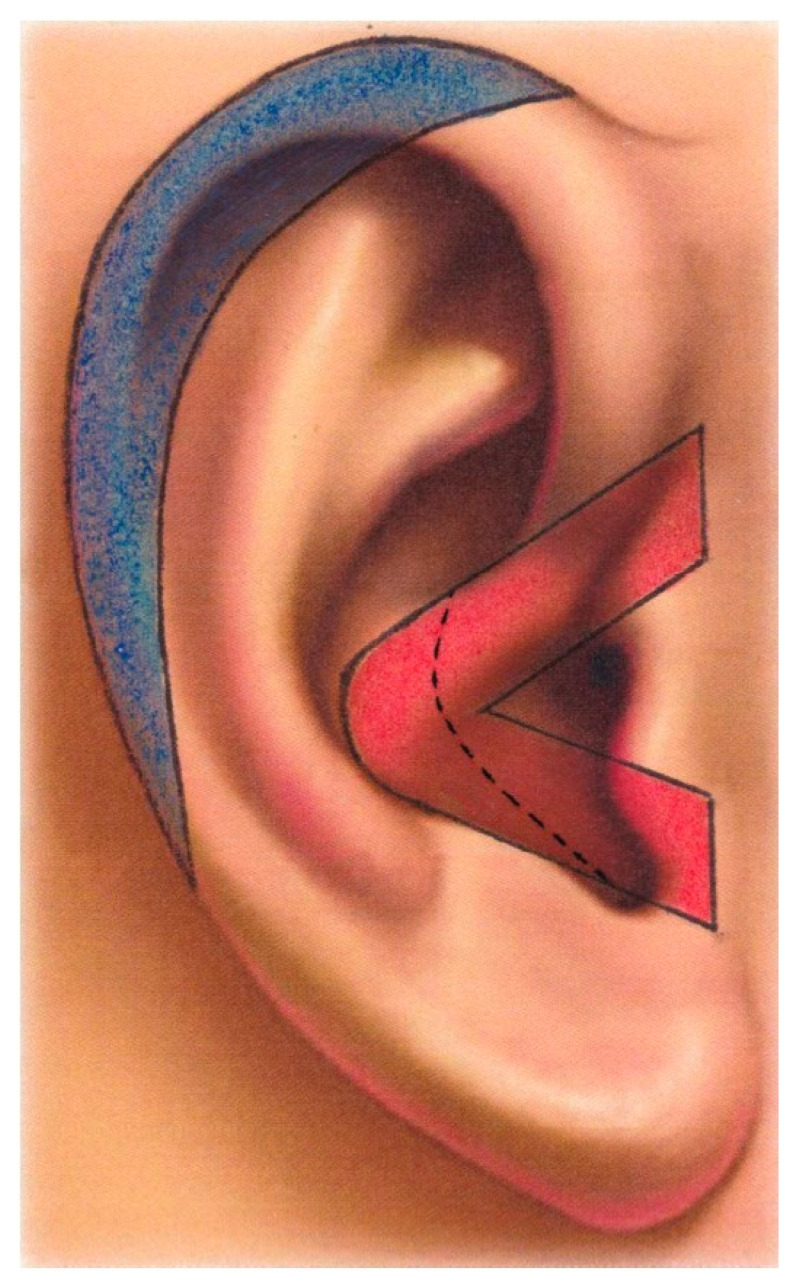
Original full-size anatomical diagram showing the sites of the embryonic fusion planes of the auricle according to Streeter, Wood-Jones, Park, Porter, and Minoux. The hyoid–mandibular fusion plane (HM–FP) is featured in red and the free ear fold-hyoid fusion plane (FEFH–FP) in blue. Reprinted with permission from Ref. [17]. 2018, SAGE Publications.

**Table 1 ijms-25-08452-t001:** Distribution of the pertinent references within the five main research domains on the dysembryogenetic pathogenesis of BCC.

Main Research Domains	References
Correlation between the topographic distribution of BCC and the macroscopic embryology	5–14, 17, 18, 53–60, 63, 64, 74
Correlation between BCC and the microscopic embryology	19–25, 27–29, 62, 65, 66
The genetic BCC	31–33
Correlation between BCC and the hair follicle	26, 30, 37–46, 52
Correlation between BCC and the molecular embryology	34–36, 61, 67–72

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
