# Peer review of "Dysembryogenetic Pathogenesis of Basal Cell Carcinoma: The Evidence to Date"

_ijms, 2024, doi:10.3390/ijms25158452_

Round 1

Reviewer 1 Report

Comments and Suggestions for Authors

the manuscript explain the dysembryogenetic pathogenesis evidence of BCC

No clear conclusion can be made as data of BCC distribution are missing.

reactivation of hedgehog inhibitor  is associated both to hereditary and sporadic BCC

the skin sensitive neve fibres express the hedgehog ligand

Author Response

Comments 1:

No clear conclusion can be made as data of BCC distribution are missing.

Response 1.

Data on BCC distribution has been included in the revised manuscript.

Reviewer 2 Report

Comments and Suggestions for Authors

The manuscript “Dysembryogenetic pathogenesis of basal cell carcinoma: the evidence to date“  could be a valuable article which presents useful data on this topic.

  As known, BCC is a sort of unique tumour due to its combined peculiar histological features and clinical behaviour, where a potential correlation between the onset of BCC and a dysembryogenetic process has long been hypothesised.  Thus, the authors conducted the analysis of the dysembryogenetic pathogenesis of BCC  and included selective investigation of PubMed-indexed publications supporting this theory retrieved 49 selected articles published between 1901-2022. According to their analysis of the literature review, 5 main research domains on the dysembryogenetic pathogenesis of BCC were identified:  (1) correlation between the topographic distribution of BCC and the macroscopic embryology, (2)  correlation between BCC and the microscopic embryology, (3) the genetic BCC, (4) correlation 24 between BCC and the hair follicle and (5) correlation between BCC and the molecular embryology.

 Finally, according to their research, a large amount of data from microscopic and molecular research consistently supports the hypothesis of a dysembryogenetic pathogenesis of BCC and such evidence is promoting advances in the clinical management of BCC as innovative targeted molecular therapies on an immune  modulating basis are being developed.

- The manuscript presents useful data that can be significant for the knowledge of BCC, but is not fully presented and interpreted in a scientifically significant way.

- What about statistical analysis of the published data from the studies?

- Bearing in mind the high level of this journal, the manuscript must be better elaborated and processed to be at a higher scientific level.

 - The manuscript lacks tables and pictures that would improve the understanding and explanation of the text. So, it could be useful to include an additional table which presents prominent study results and appropriate citing references. Thus, this presentation could be welcome to present key findings in this topic to the readers.

 - Has the analysis/research already been processed by other authors? If yes, then it should be commented. If not, then it should be stated that it is the first study with this approach to the topic.

Author Response

Comments 1:

The manuscript presents useful data that can be significant for the knowledge of BCC, but is not fully presented and interpreted in a scientifically significant way.

What about statistical analysis of the published data from the studies?

Bearing in mind the high level of this journal, the manuscript must be better elaborated and processed to be at a higher scientific level.

Response 1:

The statistical analysis of the data from the published studies, where available, has been included in the revision.

 Comments 2:

The manuscript lacks tables and pictures that would improve the understanding and explanation of the text. So, it could be useful to include an additional table which presents prominent study results and appropriate citing references. Thus, this presentation could be welcome to present key findings in this topic to the readers.

Response 2:

A comprehensive table including the prominent studies organized per main research domains on the dysembryogenetic pathogenesis of BCC has been provided.

Pictures showing the sites of embryological fusion planes of the head and neck where BCC preferentially distribute have been provided, too.

Comments 3:

Has the analysis/research already been processed by other authors? If yes, then it should be commented. If not, then it should be stated that it is the first study with this approach to the topic.

Response 3:

Several review articles on the different pathogenetic hypotheses for BCC have been reported in the literature. However, to our knowledge, ours is the first comprehensive report on the overall evidence supporting the bond between BCC and embryology.

Reviewer 3 Report

Comments and Suggestions for Authors

The manuscript “Dysembryogenetic pathogenesis of Basal Cell Carcinoma: the evidence to date” focuses on an interesting subject, the embryogenetic basis of BCC development, with particular emphasis on Hedgehog signaling pathway. This manuscript focuses on molecular oncology, being aligned with the aims and scope of the International Journal of Molecular Sciences. However, the manuscript is lacking quality in several aspects. Please refer to my comments below.

>  In the Abstract and keywords, the authors do not mention the Hedghog signaling pathway, despite being a major topic of the manuscript.

> In the Keywords, I suggest that the authors exclude “embryo” since this is not referred to in the text. Also, please include “Hedgehog signaling pathway”.

> In the Introduction, I suggest that the authors write about the risk factors and the current therapeutic management strategies for BCC since the manuscript highlights the advances in clinical management for this disease.

> Information in lines 32-37 is lacking reference(s). Please include.

> Lines 39-41: These sentences were taken “as is” from the original publication. Please rephrase accordingly.

> Among the analysed articles, there are several ones written in languages other than Italian or English (Polish, German, French). How did the authors retrieved/interpreted the information when not written in your native language or in English? Please include this in the Methods.

> Could the authors please detail what was the main goal of including the ChatGPT text? The authors do not expand on this subject and the results are only from the literature search. If the authors find it appropriate, it could be included in the Discussion or Conclusion a paragraph regarding AI role in improving BCC research. Could AI-driven systems provide accurate and up-to-date information and help researchers direct their work and aid clinicians in decision-making?

> The Results section is quite minimalist. I suggest that the authors include a Table with the 5 domains and what papers (from the 49 analysed) that had that information. In addition, these 5 domains could be used in the Discussion for a better flow of information.  

> Line 176: “The study of 2014 was repeated…”. Please add the reference of the 2014’s study.

> Lines 184-185: “…was confirmed in 1978 by novel research using both electron microscopy [22] and stem cell cultures [23].” Please rephrase the sentence. It seems that in 1978 stem cell cultures were also performed.

> For the Hedgehog signaling pathway, I suggest that the authors include information in the Introduction section since it is the major focus of the manuscript. In the current version, the reader is only briefly introduced to this key pathway in the AI-derived information and in later Discussion.

> To further improve the manuscript and enrich the Discussion section, it would be interesting to include information on the role that UV radiation may have in the abnormal activation of the Hedgehog signaling pathway. If appropriate, please consider including some text regarding this.

> In line 215, bulge stem cells are also mentioned in line 186 and here it is written that BCC does not derive from these cells. Please comment.

> In lines 224-226. The references 46-48 indicate several SMO inhibitors. Please specify the targeted therapy(ies) towards BCC that is(are) currently under development.

> Please include the years of approval of Vismodegib and Sonidegib.

> Lines 239-241. The Hedgehog pathway is highly evidenced as having a role in BCC development and pathogenesis. Targeted therapies are based or are being researched considering this pathway. What other advances in clinical management do the authors suggest from the last sentence in the Conclusion section? How can the further understanding of the dysembriogenesis translate into better clinical practices and the development of more innovative therapeutic strategies? 

Comments on the Quality of English Language

The English language needs to be revised.

Author Response

Comments 1:

In the Abstract and keywords, the authors do not mention the Hedghog signaling pathway, despite being a major topic of the manuscript.

Response 1:

Hedghog signaling pathway has now been quoted both in the abstract and the keywords.

Comments 2:

In the Keywords, I suggest that the authors exclude “embryo” since this is not referred to in the text. Also, please include “Hedgehog signalling pathway”.

Response 2:

The keywords have been revised accordingly.

Comments 3:

In the Introduction, I suggest that the authors write about the risk factors and the current therapeutic management strategies for BCC since the manuscript highlights the advances in clinical management for this disease.

Response 3:

The Introduction has been amended and the risk factors and the current therapeutic management strategies for BCC have been introduced.

Comments 4:

Information in lines 32-37 is lacking reference(s). Please include.

Response 4:

The missing references have been included.

Comments 5:

Lines 39-41: These sentences were taken “as is” from the original publication. Please rephrase accordingly.

Response 5:

The sentences from the original publication have been reduced to one.

Comments 6:

Among the analysed articles, there are several ones written in languages other than Italian or English (Polish, German, French). How did the authors retrieved/interpreted the information when not written in your native language or in English? Please include this in the Methods.

Response 6:

A few articles in languages other than English (French, German, Polish) were included in this review - the Authors being fluent in both French and German. The only retrieved article in Polish was translated to the Authors by a Polish native-speaker.

Comments 7:

Could the authors please detail what was the main goal of including the ChatGPT text? The authors do not expand on this subject and the results are only from the literature search. If the authors find it appropriate, it could be included in the Discussion or Conclusion a paragraph regarding AI role in improving BCC research. Could AI-driven systems provide accurate and up-to-date information and help researchers direct their work and aid clinicians in decision-making?

Response 7:

The role of the OpenAI ChatGPT chatbot-based search used in this review article has been discussed in the Materials and Methods section.

Comments 8:

The Results section is quite minimalist. I suggest that the authors include a Table with the 5 domains and what papers (from the 49 analysed) that had that information. In addition, these 5 domains could be used in the Discussion for a better flow of information.

Response 8:

A comprehensive table including the relevant articles within the 5 research domains on the dysembryogenetic pathogenesis of BCC has been provided. The Discussion has been entirely re-written and re-organized according to the main research domains on the dysembryogenetic pathogenesis of BCC.

Comments 9:

Line 176: “The study of 2014 was repeated…”. Please add the reference of the 2014’s study.

Response 9:

The reference has been added.

Comments 10:

Lines 184-185: “…was confirmed in 1978 by novel research using both electron microscopy [22] and stem cell cultures [23].” Please rephrase the sentence. It seems that in 1978 stem cell cultures were also performed.

Response 10:

The sentence has been re-phrased.

Comments 11:

For the Hedgehog signalling pathway, I suggest that the authors include information in the Introduction section since it is the major focus of the manuscript. In the current version, the reader is only briefly introduced to this key pathway in the AI-derived information and in later Discussion.

Response 11:

Information on the Hedgehog signalling pathway has been included in the Introduction and widely commented on in the Discussion.

Comments 12:

To further improve the manuscript and enrich the Discussion section, it would be interesting to include information on the role that UV radiation may have in the abnormal activation of the Hedgehog signaling pathway. If appropriate, please consider including some text regarding this.

Response 12:

Information on the relationship between the Hedgehog signalling pathway and UV radiation has been included in the Discussion.

Comments 13:

In line 215, bulge stem cells are also mentioned in line 186 and here it is written that BCC does not derive from these cells. Please comment.

Response 13:

The origin of the different types of BCC from different stem cell clusters of the hair follicle and similar sites within the interfollicular epidermis has been discussed with supporting evidence.

Comments 14:

In lines 224-226. The references 46-48 indicate several SMO inhibitors. Please specify the targeted therapy(ies) towards BCC that is(are) currently under development.

Response 14:

The targeted therapies towards BCC that are currently under development have been reported and discussed.

Comments 15:

Please include the years of approval of Vismodegib and Sonidegib.

Response 15:

The years of approval of Vismodegib and Sonidegib have been reported.

Comments 16:

Lines 239-241. The Hedgehog pathway is highly evidenced as having a role in BCC development and pathogenesis. Targeted therapies are based or are being researched considering this pathway. What other advances in clinical management do the authors suggest from the last sentence in the Conclusion section? How can the further understanding of the dysembriogenesis translate into better clinical practices and the development of more innovative therapeutic strategies?

Response 16:

The potential forthcoming progress in clinical management of BCC and the potential translational applications from the further understanding of the dysembryogenetic pathogenesis have been commented on in the Conclusion section.

Round 2

Reviewer 3 Report

Comments and Suggestions for Authors

The authors have addressed the questions and suggestions: Nevertheless, the manuscript still requires improvments.

> The relevance of AI-based research is still not fully explained. As I mentioned in the previous report, the authors could further discuss the potential AI role in improving BCC genesis and treatment research, otherwise the results on ChatGPT search are simply an adornment and do not bring novelty.

>The authors have added Table 1, as suggested by the Reviewer. However, the five domains should be displayed in one column, and on the other column the respective reference number should be added (only the reference number, not the entire citation).

> In lines 289-290, the authors introduce Bmi-1 and GPR49, but do not explain what these are. Please correct accordingly.

> Please revise all abbreviations. If it has been defined in the text, please use the abbreviation and not the full description. For example, the authors mention TD-derived lesions and only a few sentences later define TD as “touche domes”. Please also confirm if it is touche or touch. Other example is SMO: it first appears in line 107, and only in line 343 is defined the abbreviation.

> In the first two paragraphs of Introduction, in  lines 222-223 and lines 272-282, please add reference(s).

> In Figures and Tables please add only the reference number and not the complete citation.

> Please note that references “Moore MH, 240 David DJ, Cooter RD. Hairline indicators of craniofacial clefts. Plast Reconstr Surg. 1988;82:589–593” and “Tessier P. Anatomical classification facial, craniofacial and 236 latero-facial clefts. J Maxillofac Surg. 1976;4:69–92” are not listed in bibliography. Please correct.

Comments on the Quality of English Language

English language and grammar require revision.

Author Response

The Authors wish to thank the Reviewers for their valuable time spent in reviewing the manuscript and for their much appreciated suggestions that significantly further improved the quality of the manuscript.

All the changes in the manuscript have been highlighted in red text.

A point after point response to the Reviewers’ remarks is here provided. 

Comments 1:

The relevance of AI-based research is still not fully explained. As I mentioned in the previous report, the authors could further discuss the potential AI role in improving BCC genesis and treatment research, otherwise the results on ChatGPT search are simply an adornment and do not bring novelty.

Response 1.

The relevance and the role of AI-based research has been extensively discussed and commented on in the Discussion section.

Comments 2:

The authors have added Table 1, as suggested by the Reviewer. However, the five domains should be displayed in one column, and on the other column the respective reference number should be added (only the reference number, not the entire citation).

Response 2:

Table 1 has been amended according to the Reviewer’s suggestions.

Comments 3:

In lines 289-290, the authors introduce Bmi-1 and GPR49, but do not explain what these are. Please correct accordingly.

Response 3:

The abbreviations in lines 289-290 have been explained.

Comments 4:

Please revise all abbreviations. If it has been defined in the text, please use the abbreviation and not the full description. For example, the authors mention TD-derived lesions and only a few sentences later define TD as “touche domes”. Please also confirm if it is touche or touch. Other example is SMO: it first appears in line 107, and only in line 343 is defined the abbreviation.

Response 4:

All the abbreviations have been revised and the spelling errors have been revised.

Comments 5:

In the first two paragraphs of Introduction, in lines 222-223 and lines 272-282, please add reference(s).

Response 5:

References have been added to the first two paragraphs of the Introduction section.

Comments 6:

In Figures and Tables please add only the reference number and not the complete citation.

Response 6:

The complete citations in figures have been replaced by the reference numbers.

Comments 7:

Please note that references “Moore MH, 240 David DJ, Cooter RD. Hairline indicators of craniofacial clefts. Plast Reconstr Surg. 1988;82:589–593” and “Tessier P. Anatomical classification facial, craniofacial and 236 latero-facial clefts. J Maxillofac Surg. 1976;4:69–92” are not listed in bibliography. Please correct.

Response 7:

These references have been included in bibliography.

Comments 8:

English language and grammar require revision.

Response 8:

A complete language revision has been carried out by native English mother language experts and their contribution has been formally acknowledged.